# Subspace Networks: Scaling Decentralized Training with Communication-Efficient Model Parallelism

**Sameera Ramasinghe**    **Ajanthan Thalaiyasingam**    **Gil Avraham**    **Yan Zuo**

**Alexander Long**

Pluralis Research

## Abstract

Scaling models has led to significant advancements in deep learning, but training these models in decentralized settings remains challenging due to communication bottlenecks. While existing compression techniques are effective in data-parallel, they do not extend to model parallelism. Unlike data-parallel training, where weight gradients are exchanged, model-parallel requires compressing activations and activation gradients as they propagate through layers, accumulating compression errors. We propose a novel compression algorithm that compresses both forward and backward passes, enabling up to $99\%$ compression with no convergence degradation with negligible memory/compute overhead. By leveraging a recursive structure in transformer networks, we predefine a low-dimensional subspace to confine the activations and gradients, allowing full reconstruction in subsequent layers. Our method achieves up to $100\times$ improvement in communication efficiency and enables training billion-parameter-scale models over low-end GPUs connected via consumer-grade internet speeds as low as 80Mbps, matching the convergence of centralized datacenter systems with 100Gbps connections with model parallel.

## 1 Introduction

Scaling models and datasets has been pivotal in driving deep learning advancements, with model sizes expanding from millions of parameters [24] to billions [21, 14] and even trillions [38]. These larger models exceed the memory capacity of a single device, requiring distributed training approaches to manage computation across multiple devices.

A common solution is distributed data parallelism (DDP) [27] or its more advanced variant, fully sharded data parallelism (FSDP) [62], which distributes data across nodes while replicating the model on each device. This enables larger batch sizes and higher throughput, but constrains the model size to the memory of a single device. Model parallelism (MP) addresses this limitation by distributing parameters across devices [24, 18], enabling the training of models that surpass single-node memory constraints. MP includes tensor parallelism, which splits individual layers, and pipeline parallelism, which distributes layers across devices; the latter is the focus of this work. Modern large-scale training combines DDP and MP to achieve scalability. Despite these strategies, all approaches require the transfer of large amounts of data between devices [34], limiting training to high-performance computing clusters with fast interconnects. These infrastructures are costly and accessible only to resource-rich organizations [46, 36], creating disparities that restrict broader research and risk centralizing innovation.

Decentralized training provides an alternative by leveraging consumer-grade devices, enabling individuals with small devices to participate in large-scale model training, reducing reliance on major corporations. This approach democratizes access to large-scale model training by leveraging

39th Conference on Neural Information Processing Systems (NeurIPS 2025).

underutilized GPUs in personal computers and volunteer networks [58, 40]. However, decentralized training faces significant challenges due to limited bandwidth and high latency in heterogeneous networks [58], necessitating communication-efficient compression algorithms to minimize data transfer while preserving training performance.

Most existing communication-efficient techniques focus on DDP [28, 41, 13, 35], where *weight gradients* are computed independently on each node (for each model replica) and then compressed before peer-to-peer communication. To this end, techniques such as sparsification [54, 52, 30], quantization [53, 2, 4], and low-rank approximations [61] exploit the redundancy in weight gradients to reduce communication. However, in MP, information must be passed between *layers*, requiring the communication of *activations* and *activation gradients*. Unlike weight gradients, activations lack inherent redundancy and approximation errors accumulate across layers, leading to degraded convergence [6, 39]. These challenges prevent the straightforward application of DDP compression techniques to MP, and hence to date, MP decentralized training remains infeasible, resulting in massive slowdowns over centralized training.

To bridge this gap, we propose a novel compression algorithm tailored for MP. We show that as training progresses, weight matrices exhibit rank collapse, converging to low-rank subspaces. Thus, by *explicitly* constraining specific weight matrices to such low-rank subspaces and leveraging a recursive structure inherent in transformer networks, we demonstrate that transformer layer activations—despite their high rank—can be decomposed into a dynamic low-rank component and a static high-rank component. This decomposition enables efficient compression of information passed between layers during both the forward and backward passes, ensuring *lossless* reconstruction in subsequent layers.

We validate the practical effectiveness of our approach through extensive evaluations on billion-parameter-scale models. Our compression method enables the distribution of large-scale models across consumer-grade GPUs with internet-grade connections (80Mbps) while matching the convergence performance of centralized setups with 100 Gbps interconnects. We achieve up to $100\times$ improvement in communication efficiency without any degradation in convergence. Further, we successfully train an 8B-parameter LLaMA model [14] with layers split across four different geographical regions, connected via the internet, and achieve convergence on par with baselines utilizing datacenter-grade connections. By addressing critical limitations in decentralized training, our method intend to remove significant barriers to scaling large models in resource-constrained environments, democratizing access to large-scale deep learning.

## 2  Related works

**Decentralized training**   involves a group of *autonomous* devices (*i.e.,* no central orchestrator) collaborating to train a large-scale model by leveraging MP/DDP methods. These devices, often geographically distributed and heterogeneous, are connected via networks with varying delays and bandwidth constraints. Key advancements in this area encompass both theoretical insights [28, 23, 22] as well as practical approaches [42, 11]. Despite the progress, current efforts predominantly focus on DDP [28, 23, 22, 11], which hinders scaling models beyond the memory capacity of local nodes. SWARM parallelism [40] and Tasklets [58], treat the problem as a scheduling challenge, with the former addressing the stochasticity inherent in decentralized cluster settings. However, all methods to date still face significant scalability challenges due to communication bottlenecks inherent in MP. Our method overcomes this limitation by introducing an effective communication compression technique for MP (specifically pipeline parallel), mitigating a major barrier in scaling decentralized training.

**Communication compression**   accelerates distributed training over bandwidth-limited networks by reducing data transfers. Key strategies include **sparsification**, which transmits only significant parameter updates [54, 52, 30]; **quantization**, which lowers communication by reducing parameter precision [10, 53, 2, 4, 19, 45, 55]; and **low-rank projection**, which compresses gradients via projection onto lower-dimensional subspaces [61, 47]. While successful in data-parallel settings (DDP), these techniques face difficulties in model-parallel (MP) setups, including error accumulation across layers and unstructured activations [6, 39], causing degraded convergence. Recent work by [50] proposed low-rank MP communication, but required significant architectural changes, preventing training from scratch. In contrast, our approach involves minimal initialization changes without architectural modifications, enabling efficient MP training from scratch with improved scalability.

# 3 Transformer block

We provide a brief exposition of the transformer block and proceed to describe the proposed compression method. Let the input to the $l^{\text{th}}$ layer be $\mathbf{X}^l \in \mathbb{R}^{b \times n \times d}$, where $b$, $n$, and $d$ are the batch size, sequence length, and embedding dimension, respectively. Given the weight matrices of each attention head $h$ as $\mathbf{W}_{Q,h}^l, \mathbf{W}_{K,h}^l, \mathbf{W}_{V,h}^l \in \mathbb{R}^{d \times d_H}$, with $d_H = d/H$ where $H$ is the number of attention heads, the following computations are performed for each attention head: $\mathbf{X}_{Q,h}^l = \mathbf{X}^l \mathbf{W}_{Q,h}^l$, $\mathbf{X}_{K,h}^l = \mathbf{X}^l \mathbf{W}_{K,h}^l$, $\mathbf{X}_{V,h}^l = \mathbf{X}^l \mathbf{W}_{V,h}^l$. The rest of the computations are as follows:

$$\mathbf{X}_{\text{head, h}}^l = f_{\text{softmax}} \left( \frac{\mathbf{X}_{Q,h}^l \mathbf{X}_{K,h}^{l\top}}{\sqrt{d_H}} \right) \mathbf{X}_{V,h}^l \tag{1}$$

$$\begin{aligned} \mathbf{X}_{\text{concat}}^l &= [\mathbf{X}_{\text{head, 1}}^l, \ldots, \mathbf{X}_{\text{head, H}}^l] \\ \mathbf{X}_{\text{attn}}^l &= \mathbf{X}_{\text{concat}}^l \mathbf{W}_{p_1}^l + \mathbf{X}^l \\ \mathbf{X}_{\text{hidden}}^l &= f_{\text{relu}}(\mathbf{X}_{\text{attn}}^l \mathbf{W}_1^l) \\ \mathbf{X}^{l+1} &= \mathbf{X}_{\text{hidden}}^l \mathbf{W}_{p_2}^l + \mathbf{X}_{\text{attn}}^l \end{aligned} \tag{2}$$

where $\mathbf{W}_{p_1}^l \in \mathbb{R}^{d \times d}$, $\mathbf{W}_1^l \in \mathbb{R}^{d \times d_{\text{ff}}}$, and $\mathbf{W}_{p_2}^l \in \mathbb{R}^{d_{\text{ff}} \times d}$. We omit the layer norms for brevity, which does not affect any of our derivations. $d_{\text{ff}}$ is usually an integer multiple of $d$. We will refer to $\mathbf{W}_{p_2}^l$ and $\mathbf{W}_{p_1}^l$ as *projection matrices* from here onward.

# 4 Subspace networks

## 4.1 Rank collapse of projection matrices

We leverage the observation that weight gradients of projection matrices inherently reside in a low-dimensional subspace, as widely reported across various architectures [26, 47, 51, 59, 60, 7, 56, 16] (see Appendix E.1 for empirical validation). Together with AdamW's decoupled weight decay—which suppresses negligible gradient components—this drives projection matrices to converge toward a subspace spanned by dominant gradients, resulting in effectively low-rank structures. Formal treatment is provided in Statements 5.2, 5.3, and their associated theorems.

To validate this phenomenon, we train an 8-layer, 2B-parameter model on WikiText [33] and track the stable rank of its projection layers during training (hidden dimension of 4096 and a context length of 2048. Stable rank is computed as $\sum_i \sigma_i^2 / \max_i(\sigma_i^2)$, where $\sigma_i$ denotes the $i^{\text{th}}$ singular value. As shown in Fig. 1, the stable rank of projection matrices sharply declines, consistent with our theory. This structure, combined with the recursive nature of transformers, allows us to design a nearly-lossless low-rank compression scheme. While prior works observe similar rank collapse in self-attention matrices [12, 1, 43], we focus specifically on projection matrices as the basis for compression. We also validate this using official checkpoints of fully-trained large scale models (Appendix J).

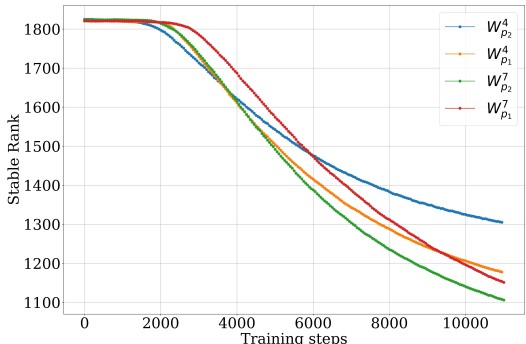

Figure 1: **Rank collapse in projection matrices.** Consistent with Statements 5.2 and 5.3 (and Theorems E.2, G.1 - Appendix), we empirically observe a natural rank collapse in the projection matrices (of non-compressed models). Shown is an 8-layer, 2B-parameter model, with the stable (effective) ranks of the projection matrices for the $4^{\text{th}}$ (middle) and $7^{\text{th}}$ (penultimate) layers plotted over training steps.

## 4.2 Investigating the activation structure

We utilize the natural rank collapse we discussed thus far for compression. Observing the transformer block in Eq. 1 and 2, we identify that a recursive structure emerges for layer outputs due to the skip connections:

$$\mathbf{X}^{l+1} = \mathbf{X}^l_{\text{hidden}}\mathbf{W}^l_{p_2} + \mathbf{X}^l_{\text{concat}}\mathbf{W}^l_{p_1} + \mathbf{X}^l. \tag{3}$$

which can be expanded as:

$$\mathbf{X}^{l+1} = \sum_{i=1}^{l}(\mathbf{X}^i_{\text{hidden}}\mathbf{W}^i_{p_2} + \mathbf{X}^i_{\text{concat}}\mathbf{W}^i_{p_1}) + \mathbf{X}^0 \tag{4}$$

Here,

$$\mathbf{X}^0 = \text{PE} + \text{TE}, \tag{5}$$

where $\text{PE}, \text{TE} \in \mathbb{R}^{b \times n \times d}$ represent the positional and token embeddings, respectively. Let $[p_1, p_2, \ldots, p_n]$ be the sequence of positional indices, and $[t_1, t_2, \ldots, t_n]$ be the corresponding token indices. We denote embedding matrices $\mathbf{P} \in \mathbb{R}^{n \times d}$ for positional embeddings and $\mathbf{T} \in \mathbb{R}^{v \times d}$ for token embeddings, where $v$ is the vocabulary length. Thus the tokens and positional indices are embedded via a lookup: $\text{PE} = \mathbf{P}[p_{1:n}, :]$ and $\text{TE} = \mathbf{T}[t_{1:n}, :]$.

Observing Eq. 4, we note that $\mathbf{X}^0 = \text{PE} + \text{TE}$ contributes as a common additive term to all layer outputs. Therefore, we consider the rank of the residual activations when PE and TE are subtracted:

$$\hat{\mathbf{X}}^{l+1} = \mathbf{X}^{l+1} - \text{PE} - \text{TE} = \sum_{i=1}^{l}(\mathbf{X}^i_{\text{hidden}}\mathbf{W}^i_{p_2} + \mathbf{X}^i_{\text{concat}}\mathbf{W}^i_{p_1}) \tag{6}$$

Recall that $\text{Row}(\mathbf{AB}) \subseteq \text{Row}(\mathbf{B})$, for any two matrices $\mathbf{A}, \mathbf{B}$ where $\text{Row}(\cdot)$ denotes the row space. Thus, it is clear that if the rows of the projection weights $(\mathbf{W}_{p_2}, \mathbf{W}_{p_1})$ up to layer $l$ span a common low-dimensional subspace, then the rows of $\hat{\mathbf{X}}^{l+1}$ is also restricted to the same subspace, since vector spaces are closed under addition.

## 4.3 Compressing the forward pass

Recall that our analysis so far indicates that the projection matrices naturally confine themselves to a smaller subspace as training progresses. Consequently, the residual activations $\hat{\mathbf{X}}^{l+1}$ are also restricted to a smaller subspace, if the union of those subspaces is low-dimensional. Based on this insight, it is intriguing to explore the feasibility of *explicitly forcing* the rows of projection matrices to vary within a common low-dimensional subspace $\mathcal{S}$, throughout training, to facilitate activation compression. As discussed in Section 4.2, this forces the rows of activation outputs $(\hat{\mathbf{X}}^{l+1})$ to span the same subspace $\mathcal{S}$. Surprisingly, we find that even with extreme low-dimensional $\mathcal{S}$, the networks can achieve almost the same convergence rates as in the unaltered ones. This allows us to significantly reduce the communication between blocks, which we show next.

Let $\text{Row}(\mathbf{W}^l_{p_2}), \text{Row}(\mathbf{W}^l_{p_1}) \subseteq \mathcal{S}$. Further, Let $\mathbf{U}_k \in \mathbb{R}^{d \times k}$ be a matrix with orthonormal columns and $\text{Col}(\mathbf{U}_k) = \mathcal{S}$. Then, the following holds:

$$\hat{\mathbf{X}}^{l+1} = \hat{\mathbf{X}}^{l+1}\mathbf{U}_k\mathbf{U}_k^\top \tag{7}$$

In other words, $\hat{X}^{l+1}$ remains unaltered by the projection, since it is already in $\mathcal{S}$. The above formulation introduces a property that can be leveraged for compression during the forward pass. Specifically, the dimensionality of $\hat{\mathbf{X}}^{l+1}\mathbf{U}_k \in \mathbb{R}^{b \times n \times k}$, is substantially smaller than that of $\hat{\mathbf{X}}^{l+1} \in \mathbb{R}^{b \times n \times d}$ since $k \ll d$. If each node in a distributed system is initialized with a shared copy of $\mathbf{U}_k$, *this matrix does not need to be transmitted repeatedly*. Instead, $\hat{\mathbf{X}}^{l+1}\mathbf{U}_k$ can be transmitted to the next node, and the original $\mathbf{X}^{l+1}$ is reconstructed as:

$$\mathbf{X}^{l+1} = \hat{\mathbf{X}}^{l+1}\mathbf{U}_k^\top + \text{PE} + \text{TE}.$$

This approach ensures exact recovery of $\mathbf{X}^{l+1}$ without approximation.

### 4.3.1 Decomposition of high-rank components

For practical utilization of above approach, we still need to subtract PE and TE from $\mathbf{X}^{l+1}$ to compute $\hat{\mathbf{X}}^{l+1}$ (and add them back in the next layer). While PE is deterministic and can be computed locally within each node, TE varies depending on the batch, making it impossible to do so.

One potential solution is restricting the embedding table $\mathbf{T}$ also to $\mathcal{S}$. However, we observed that this degrades network performance due to severely limiting the representation capacity of the token embeddings. Instead, we propose modeling TE as a composition of a fixed high-rank component and a trainable low-rank component:

$$\text{TE} = \mathbf{T}_{\text{fixed}}[t_{1:n}, :] + \mathbf{T}_{\mathcal{S}}[t_{1:n}, :],$$

where we obtain a trainable low rank embedding table $\mathbf{T}_{\mathcal{S}} = \mathbf{T}_{\text{fixed}} \mathbf{U}_k \mathbf{U}_k^T$. At the beginning of training, $\mathbf{T}_{\text{fixed}}$ is transmitted to all nodes and stored. During the forward pass, we compress the activations as:

$$
\begin{aligned}
\mathbf{X}_{\text{compressed}}^{l+1} &= (\hat{\mathbf{X}}^{l+1} + \mathbf{T}_{\mathcal{S}}[t_{1:n}, :])\mathbf{U}_k \\
&= (\mathbf{X}^{l+1} - \text{PE} - \mathbf{T}_{\text{fixed}}[t_{1:n}, :])\mathbf{U}_k,
\end{aligned}
\tag{8}
$$

Note that in Eq. 8 both PE and $\mathbf{T}_{\text{fixed}}[t_{1:n}, :]$—which are generally high-rank—are subtracted from $X^{l+1}$ so the remaining $(\mathbf{X}^{l+1} - \text{PE} - \mathbf{T}_{\text{fixed}}[t_{1:n}, :])$ is already in $\mathcal{S}$ and is low-rank. Further, since $\text{Row}(\mathbf{T}_{\mathcal{S}}[t_{1:n}, :]) \subseteq \text{Col}(\mathbf{U}_k)$, it is implicitly captured in the compressed activations $\mathbf{X}_{\text{compressed}}^{l+1}$. Reconstruction at the next node is then performed as:

$$\mathbf{X}_{\text{recovered}}^{l+1} = \mathbf{X}_{\text{compressed}}^{l+1} \mathbf{U}_k^\top + \text{PE} + \mathbf{T}_{\text{fixed}}[t_{1:n}, :]) = \mathbf{X}^{l+1}$$

ensuring a **lossless** recovery of $\mathbf{X}^{l+1}$ while not compromising its high ranked-ness.

A natural question arises: would explicitly restricting the projection matrices to a fixed $\mathcal{S}$, instead of allowing this property to emerge organically, adversely affect convergence? Note that this is a form of constraint optimization and there are well known convergence guarantees. However, in Sec. 5, we provide a convergence proof (to at least a first-order stationary point) for completeness for the above partial projection case. Furthermore, we conduct extensive experiments across a variety of settings to empirically validate the convergence. Further, since the fixed embeddings are ephemeral, they have a negligible effect on the peak GPU memory (Appendix K)

### 4.4 Compression in backpropagation

In the previous section, we showed that constraining the rows of projection matrices to a shared low-dimensional subspace $\mathcal{S}$, coupled with decomposing the embedding table into low-rank and high-rank components, facilitates compression of activations in the forward pass. This same constraint naturally facilitates lossless gradient compression in the backward pass. Specifically, let $\nabla_L(\mathbf{X}^{l+1})$ denote the gradient of $\mathbf{X}^{l+1}$, with respect to the loss, that needs to be propagated to the previous layer. This gradient can be compressed as:

$$\left(\nabla_L(\mathbf{X}^{l+1})\right)_{\text{compressed}} = \nabla_L(\mathbf{X}^{l+1})\mathbf{U}_k \in \mathbb{R}^{b \times n \times k}, \tag{9}$$

and subsequently fully recovered in the previous layer $l$ as:

$$\left(\nabla_L(\mathbf{X}^{l+1})\right)_{\text{recovered}} = \left(\nabla_L(\mathbf{X}^{l+1})\right)_{\text{compressed}} \mathbf{U}_k^\top = \nabla_L(\mathbf{X}^{l+1}). \tag{10}$$

Remarkably, this formulation ensures that the gradient flow to the computational graph prior to $\mathbf{X}^{l+1}$ remains lossless, with no approximation error. Intuitively, after backpropagation through the parameter matrix $\mathbf{W}_{p_2}$, the gradient has the form $\nabla_L(\mathbf{X}^{l+1})\mathbf{W}_{p_2}^\top$. Because $\text{Row}(\mathbf{W}_{p_2}) \subseteq \text{Col}(\mathbf{U}_k) = \mathcal{S}$, the projection $\nabla_L(\mathbf{X}^{l+1})\mathbf{U}_k \mathbf{U}_k^\top \mathbf{W}_{p_2}^\top$ does not alter the resulting gradient flow. Full derivation is provided in Appendix C.

## 4.5 Subspace updates using Grassmann manifold

We observe that restricting the column spaces of projection layers to a fixed subspace, even at high-ratios, is able to maintain surprisingly adequate convergence. To further improve the convergence, we allow the subspaces to slowly drift. To align the subspace with the gradient directions, we minimize the norm of the gradient components that lie outside the subspace, as measured at the last Transformer layer. Let $\nabla_L(\mathbf{X}_t^{\text{final}}) \in \mathbb{R}^{b \times n \times d}$ denote the activation gradients at the last compressed transformer layer. The leftover gradient, which lies outside the subspace, is $\hat{\mathbf{X}}_t^{\text{final}} = \nabla_L(\mathbf{X}_t^{\text{final}})(\mathbf{I} - \mathbf{U}_k\mathbf{U}_k^\top)$, where $\mathbf{I} - \mathbf{U}_k\mathbf{U}_k^\top$ is the projection operator onto $\mathcal{S}^\perp$. We accumulate $\hat{\mathbf{X}}_t^{\text{final}}$ over $K$ iterations to obtain the metric $\mathcal{L}_{\text{Grassmann}} = \frac{1}{K}\sum_{t=k}^{k+K}\|\hat{\mathbf{X}}_t^{\text{final}}\|_F^2$, where $\|\cdot\|_F$ is the Frobenius norm. We aim to minimize $\mathcal{L}_{\text{Grassmann}}$ over all possible $\mathbf{U}_k$.

A straightforward way to minimize $\mathcal{L}_{\text{Grassmann}}$ is by performing SVD on it and updating the subspace using the left singular vectors. However, abrupt changes to the subspace can disrupt convergence. Thus, we perform smooth updates by taking steps on the Grassmann manifold. The Grassmann manifold $\mathcal{G}(k, n)$ is the set of all $k$-dimensional subspaces of $\mathbb{R}^n$. A point on $\mathcal{G}(k, n)$ is represented by an orthonormal matrix $\mathbf{U}_k \in \mathbb{R}^{n \times k}$, where the columns of $\mathbf{U}_k$ form a basis for the subspace. Thus, defining $\mathcal{S}$ as a point on the Grassmann manifold enables taking smooth steps on the manifold. To minimize $\mathcal{L}_{\text{Grassmann}}$, we employ gradient descent on $\mathcal{G}(k, n)$. First, the Euclidean gradient of $\mathcal{L}_{\text{Grassmann}}$ with respect to $\mathbf{U}_k$, denoted $\nabla_{\mathcal{L}_{\text{Grassmann}}}(\mathbf{U}_k)$, is projected onto the tangent space to obtain the Riemannian gradient:

$$(\nabla_\mathcal{L}\mathbf{U}_k)_{\text{tangent}} = \nabla_{\mathcal{L}_{\text{Grassmann}}}(\mathbf{U}_k) - \mathbf{U}_k\mathbf{U}_k^\top \nabla_{\mathcal{L}_{\text{Grassmann}}}(\mathbf{U}_k). \tag{11}$$

Then, we perform a gradient descent step $\mathbf{U}_k^{\text{new}} = \mathbf{U}_k - \eta\,(\nabla_\mathcal{L}\mathbf{U}_k)_{\text{tangent}}$ where $\eta$ is the step size. Then, to map $\mathbf{U}_k^{\text{new}}$ back to the manifold, we apply a retraction by orthonormalizing the columns of $\mathbf{U}_k^{\text{new}}$ using QR decomposition $\mathbf{U}_k^{\text{new}}, \mathbf{R} = \text{QR}(\mathbf{U}_k^{\text{new}})$, where $\text{QR}(\cdot)$ denotes the QR decomposition. In practice, we perform this subspace update on $\mathbf{U}_k$ very infrequently (per every $500$ iterations), and transmit to all the layers.

# 5 Theoretical insights

This section provides several theoretical insights for the proposed method. We structure our analysis into four key statements (and put the corresponding formal theorems in the Appendix). The first statement indicates that if there is a lossy compression between the layers, as the model depth increases, the approximation error of compression can grow exponentially. The second and third statements demonstrate that, even in an uncompressed network, if the weight gradients are confined to a particular subspace, the weight matrices also naturally converge to a low-dimensional subspace with AdamW. The fourth observation establishes that explicitly enforcing a subset of weights onto a low-dimensional subspace does not harm convergence.

---

**Statement 5.1**

If the compression of activations and activation gradients between layers in a model-parallel setting introduces approximation errors, these errors can accumulate exponentially with increasing depth, provided the weight and activation norms are sufficiently large. Refer to Theorem D.1 for a formal proof.

---

The above result suggests that extending compression techniques from DDP (which are lossy) to MP (which requires compressing information passed between adjacent layers) leads to the accumulation of approximation errors. This occurs because the compression at one layer directly impacts downstream layers, a phenomenon not present in DDP. Additionally, the lack of exploitable structure in activations and activation gradients [6, 39] typically results in larger approximation errors compared to the gradients of weights. This limitation makes such compression methods unsuitable for MP in large-scale models.

> **Statement 5.2**
>
> If the gradients of a particular weight matrix in a network are constrained to a fixed subspace, then under AdamW, the weight *updates* asymptotically converge to that subspace over a sufficiently large number of training steps. For a formal proof, see Theorem E.2.

This provides a critical insight: if the gradients of an unconstrained network predominantly lie within a specific low-dimensional subspace $\mathcal{S}$—a property we empirically validate (see Appendix C)—then the AdamW optimizer asymptotically restricts updates outside of $\mathcal{S}$. While this behavior is straightforward for vanilla stochastic gradient descent (SGD), it is non-trivial for AdamW due to its adaptive learning rate mechanism.

> **Statement 5.3**
>
> If Statement 5.2 holds, then the corresponding weight matrices asymptotically converge to the same subspace, irrespective of their initialization. For a formal proof, see Theorem G.1.

Intuitively, this result indicates that the decoupled weight decay mechanism in AdamW systematically suppresses components of the weight matrix that receive negligible gradient updates. Consequently, the learned weights converge to a low-dimensional subspace defined by the gradient updates.

> **Statement 5.4**
>
> A network in which a subset of weights is constrained to a low-dimensional subspace converges to a first-order stationary point with a convergence rate of $O(1/T)$. For a formal proof, see Proposition H.1.

This result is a straightforward extension of the the convergence rate guarantees for constrained optimization using proximal gradient descent on non-convex functions. It shows that even when a *subset* of parameters is restricted to a lower-dimensional subspace, the standard convergence rate of $O(1/T)$ in terms of stationarity remains intact.

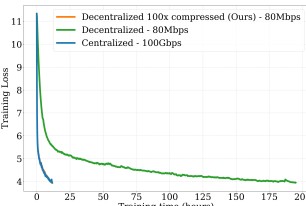 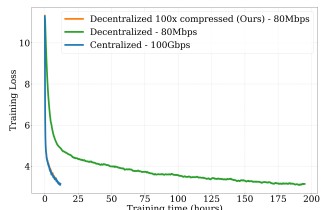 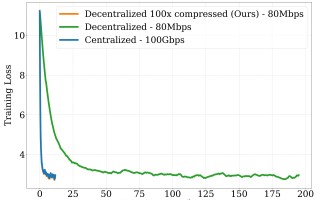

Figure 2: **Convergence in low-bandwidth settings.** From left to right: OpenWebText, WikiText, and BookCorpus. In each plot, the training curves are presented against wall-clock time for an 8-layer (2B) model. Decentralized models utilize 80Mbps connections while the centralized model has datacenter-grade 100Gbps links. Our compressed model achieves on-par convergence to the centralized model, even under a 80Mbps bandwidth budget. In contrast, the non-compressed decentralized model with 80Mbps links suffers from significantly slower convergence due to the communication bottleneck.

## 6 Experiments

### 6.1 Experimental Setup

We evaluate decoder-only models (based on Llama 3 [14]) across four large-scale datasets: WikiText (WT) [33], BookCorpus (BC) [63], OpenWebText (OWT) [15], and C4 [37]. For WT, we use the standard splits; for BC and OWT, we randomly select $10\%$ of training data as validation; for C4, due to computational constraints, we report training loss only. The base model has a context length of $1024$, embedding dimension $4096$, 24 heads, and 8 layers ($\sim$2B parameters); larger models (up to 8B parameters) are noted explicitly in ablation sections. We use a base learning rate $\eta = 3e\text{-}4$ (with warmup and linear decay), weight decay $0.01$, and batch size 32, unless otherwise specified. We

use GPipe [18] via `torch.distributed.pipelining`, integrating our compression into all but the final transformer layer.

We initialize $\mathbf{U}_k$ with isotropic Gaussian noise and set $k = 40$, achieving $100\times$ compression. Bandwidth simulations sample from $\mathcal{N}(\mathcal{B}, 0.2\mathcal{B})$ per pass, defining 'centralized' as 100Gbps or 16Gbps setups, with all others as 'decentralized'. Experiments (except the 8B Llama run on L4 GPUs with internet-based decentralized connections) use A10g GPUs (24GB VRAM) with one layer per GPU. **Our method's effectiveness increases with faster accelerators, as slower GPUs allow more computation-communication overlap.**

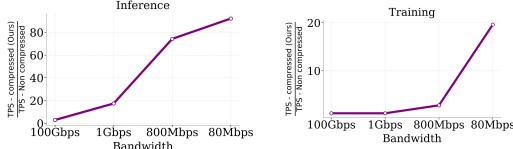

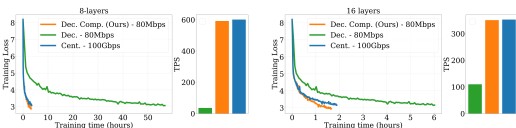

Figure 3: **Throughput gain.** Compressed models significantly improve throughput under bandwidth constraints for inference (left) and training (right). Results shown for 8-layer (2B) models.

Figure 4: **Performance vs. depth.** Our compression matches or surpasses centralized baselines as depth increases (left: 8 layers; right: 16 layers). With two layers per GPU, computational load rises, slightly narrowing the gap between centralized and decentralized models.

**Square-Cube Law.** Square-cube law [40] states that in distributed training, computation scales cubically with model size per node, while communication grows only quadratically. This partially offsets communication bottlenecks with computational overhead. Thus, $c$-times slower communication does not lead to a $c$-times slower convergence. Hence, by improving communication efficiency by $100\times$, we achieve convergence speeds comparable to 100Gbps setups, even with 80Mbps links.

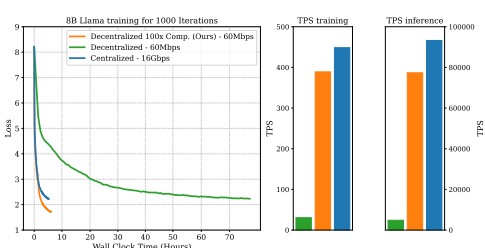

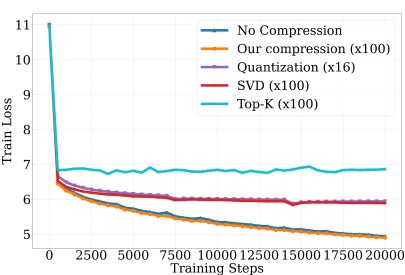

Figure 5: **Training convergence and throughput on an 8B LLaMA model.** All runs use 64 L4 GPUs distributed over 8 instances. Centralized instances reside within one region (min bandwidth 16 Gbps), while decentralized instances span 4 regions (min bandwidth 60 Mbps), highlighting pipeline parallel bottlenecks due to reduced inter-node bandwidth.

Figure 6: **Comparison against lossy compression methods.** Top-k, low-rank (SVD), and quantization fail to converge at $100\times$ compression, with quantization additionally limited by numerical precision. Our method matches the convergence rate of the uncompressed baseline.

## 6.2 Convergence in low-bandwidth settings

Our method enables training models over extremely low-bandwidth connections. We trained networks on both 80 Mbps and datacenter-grade 100 Gbps connections. Fig. 2 illustrates the train curves of an 8-layer (2B) model against wall-clock time. As expected, training over 80 Mbps links in the decentralized setting significantly degrades convergence. In contrast, with our compression, the decentralized model achieves on par convergence to the network trained over 100 Gbps connections. To demonstrate the generalization, the perplexity scores of each model over validation sets is shown in Table. 1. As evident, the decentralized model with our compression even surpasses the performance of the centralized model for the same training time. To further validate test-time performance, we

train models to convergence using the compute-optimal 1:20 model-to-token ratio from the Chinchilla scaling law [17], reaching compute-optimality at 12B training tokens with superior performance (Appendix F).

## 6.3 Throughput gain

Our compression also significantly accelerates inference. As inference requires less computation than training, bandwidth becomes the dominant bottleneck; hence, our compression yields substantial gains. Fig. 3 illustrates gains in both training and inference: at inference, we achieve almost a $100\times$ speedup at 80 Mbps. Although this advantage diminishes at higher bandwidths (e.g., 100 Gbps), we still observe about a $3\times$ improvement. **This indicates that even centralized systems benefit from reduced inference latency using our approach, which can translate into considerable cost savings with the recent trend of inference time scaling of large language models [44, 5]** A similar trend holds for training throughput as well.

Table 1: **Perplexity scores**. Models trained for 12 hours on OpenWebText (OWT), BookCorpus (BC), and WikiText (WT). Bandwidth (B/W) and tokens per second (TPS) are reported. Our method outperforms even the centralized model, achieving significantly higher TPS compared to the non-compressed decentralized baseline.

| Model | B/W | OWT↓ | BC↓ | WT↓ | TPS↑ |
|---|---|---|---|---|---|
| Dec. | 80Mbps | 925.19 | 108.85 | 601.84 | 36.12 |
| Dec. Comp. | 80Mbps | **46.75** | **17.63** | **23.01** | 592.41 |
| Cen. | 100Gbps | 47.22 | 18.35 | 23.08 | 602.57 |

## 6.4 Scaling across globally distributed GPUs

To further explore the scalability, we trained a LLaMA 8B parameter variant [14] with 2048 context length, using TorchTitan [29] on the C4 dataset across 64 L4 GPUs distributed across 8 instances. We use a pipeline parallel setup with 32 stages running in 2 FSDP dimensions, where the 32 transformer layers are distributed one layer per stage. We evaluated two environment configurations: Centralized and Decentralized. In the Centralized setting all instances were located in the same cloud region and the bandwidth spans between 16Gbps-27Gbps. For the Decentralized case the 8 instances were distributed across 4 distinct regions (North America, Europe, and Asia). Additionally, no two consecutive stages were placed in the same region for the decentralized setup, hence, bandwidth spans from 60Mbps-350Mbps. As shown in Fig. 5, our compression method in the decentralized configuration matches the wall-clock time (even slightly improving) and TPS with the centralized setting. In contrast, the decentralized setting w/o compression was $13\times$ *slower*.

## 6.5 On other communication efficient distributed training methods

Our method is tailored for model parallel compression and therefore is orthogonal and complementary to data parallel communication-reduction techniques. This enables layering our compression on top of them. To illustrate this, we present an experiment: we applied top-k and low-rank compression (powerSGD [47]) to the weight gradients (replicating a DP scenario) while simultaneously using our scheme to compress activations and activation gradients. Under an 80Mbps link between devices, we achieved a substantial increase in TPS with no loss of convergence (see Table 2).

## 6.6 Ablations

We conduct ablations on the C4 dataset to evaluate the robustness of our method. Fig. 4 compares performance across model depths. If our compression were lossy, deeper models would accumulate errors, degrading performance relative to the centralized baseline [6] (see also Theorem D.1). However, our results show that even as depth increases from 8 (2B) to 16 (3.5B) layers, convergence remains on par with centralized baselines. Further, our large-scale experiment

Table 2: **Our compression can be overlayed on DDP compression methods**.

| Method | TPS | Perplexity↓ |
|---|---|---|
| Top-K (10%) | 32 | 52.98 |
| Top-K (10%) + Ours | **599** | **52.45** |
| PowerSGD | 29 | 31.18 |
| PowerSGD + Ours | **532** | **30.26** |

(Fig. 5) confirms that 32-layer models scale ef-
fectively, demonstrating decentralized training of large models with MP for the first time. Fig. 4 also
highlights that in the 16-layer model, assigning two layers per GPU (A100, 40GB VRAM) increases
per-GPU computation, reducing bandwidth bottlenecks and narrowing the gap between decentralized
and centralized models. This validates the square-cube law [40], showing how computation-to-
communication balance impacts decentralized training. However, note that decentralized training
primarily targets low-end GPUs rather than high-end hardware. Ablations over other design choices
and the negligible memory overhead are discussed in Appendix K.

### 6.7 Comparison against lossy compressions

As per Statement 5.1, standard compression methods used in DDP do not effectively extend to
MP. We train an 8-layer model on the WikiText, comparing our compression method with TopK,
quantization, and low-rank projection. As shown in Fig. 6, with an aggressive compression rate of
$\times 100$, such compression schemes fail to converge. In contrast, our method achieves convergence on
par with the non-compressed model. Note that for quantization, the best compression rate we can
achieve is $16\times$ for 16bit precision.

## 7  Conclusion

We propose a novel compression technique that, for the first time, enables aggressive compression in
model parallel. By leveraging structured subspace constraints, we achieve up to $100\times$ communication
efficiency while preserving convergence. Our compression enhances both inference and training
efficiency, reducing latency even in centralized settings. Extensive experiments across varying model
depths, bandwidth conditions, and large-scale deployments validate the effectiveness and scalability
of our approach. Notably, we demonstrate its real-world applicability by successfully training an
8B-parameter Llama model on low-end GPUs distributed across multiple global regions, connected
solely via internet-grade (60 Mbps) links, while achieving convergence comparable to a centralized
setup. Our results open a practical pathway towards decentralized training of large-scale models
using model parallelism.

## 8  Limitations

While we provide foundational theoretical analysis, deeper insights into our method would be
beneficial. Notably, our compressed models sometimes outperform uncompressed baselines at
equal training iterations, potentially due to implicit regularization effects. A rigorous exploration of
this phenomenon remains open. Additionally, extending our analysis to advanced optimizers (e.g.,
AdamW) and deriving tighter convergence bounds would strengthen the theoretical grounding; we
leave these directions for future work.

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
