# OpenReview forum: "Subspace Networks: Scaling Decentralized Training with Communication-Efficient Model Parallelism"
_NeurIPS.cc/2025/Conference — NeurIPS 2025 poster_

### Official Review · Reviewer_VZDz · 2025-06-30

**Clarity:** 2
**Significance:** 4
**Originality:** 3
**Rating:** 4
**Confidence:** 4

**Summary:**

This paper presents a communication compression algorithm designed to overcome bottlenecks in decentralized, model-parallel training of large-scale transformer models. The core idea is to leverage the natural rank collapse observed in transformer projection matrices by explicitly constraining these weights to a predefined low-dimensional subspace shared across devices. This constraint enables the nearly lossless compression and reconstruction of activations and gradients during both forward and backward passes, thereby avoiding the error accumulation that typically plagues compression techniques in MP settings. The authors empirically demonstrate that their method achieves up to 100x communication efficiency, enabling billion-parameter models to be trained over slow, consumer-grade internet connections while matching the convergence performance of centralized systems with high-speed interconnects.

**Questions:**

1. The proposed method focuses on compressing the projection matrices by constraining their row spaces. Is this approach orthogonal to, and potentially compatible with, other compression techniques? For example, could Subspace Networks be combined with methods that induce low-rank structure in self-attention matrices  or with other techniques used in DDP to achieve even greater communication savings?
2. Regarding Statement 5.4, which guarantees convergence to a first-order stationary point for the constrained network: How does this constrained optimum relate to the stationary points of the original, unconstrained problem? Could the subspace constraint potentially prevent the model from reaching a better (e.g., lower loss) stationary point that exists outside the predefined subspace?
3. In Figure 4, the 16-layer model appears to reach a similar training loss as the 8-layer model in much fewer training time. This is a notable result, as the 16-layer model is significantly larger and presumably requires more computation per step. Could the authors provide insight into why the larger model does not incur a proportionally longer training time to reach convergence in this setting?

Minor suggestions:

1. I would recommend increasing the font size for axis labels and legends in several figures, particularly Figures 1 and 4. The current size makes the details difficult to parse.
2. There is a minor formatting inconsistency in the plot titles within Figure 4 ('8-layers' vs. '16 layers').
3. In Figure 2, the convergence curves for the proposed compressed model and the centralized baseline are highly overlapping. It makes the plots difficult to visually distinguish.

**Ethical Concerns:**

["NO or VERY MINOR ethics concerns only"]

**Final Justification:**

I think this is a technically good paper. The design are well-motivated with solid theorectical foundation and are verified through experiments.  The main problem with this manuscript lies in the presentation. The logical completeness of the writing can be further enhanced. However, the minor flaws do not overshadow the overall excellence, so I have decided to increase my score.

**Limitations:**

yes

**Quality:**

3

**Strengths And Weaknesses:**

Strengths:

1. The paper tackles the critical and highly relevant problem of communication efficiency in model-parallel training. By proposing a nearly-lossless compression method, this work has significant practical implications.
2. The central insight is novel and well-motivated. The authors convincingly demonstrate—both theoretically and empirically—that key transformer weight matrices can be constrained to a shared low-dimensional subspace throughout training without degrading model convergence.
3. The empirical results are a standout feature. Most notably, experiments show that a 2B-parameter model trained with the proposed compression over a 80Mbps connection achieves convergence on par with a baseline on a 100Gbps link.
4. The methodology is well-grounded in theory, built upon the observation of rank collapse in projection matrices. The authors provide a solid theoretical framework, including proofs that justify the method's lossless nature and formal guarantees that the constrained network converges.



Weaknesses:

1. The paper lacks a clear, explicit description of the mechanism used to enforce the subspace constraint on the projection matrices. While it is stated that the rows of these matrices lie within the common subspace, the paper does not specify the detailed implementation.
2. A crucial ablation study on the hyperparameter k (the subspace dimension) is notably absent. The experiments use a fixed k=40, but an analysis of the trade-off between compression ratio and model performance is essential for understanding the method's robustness and practical application.
3. While a mechanism to update the shared subspace using the Grassmann manifold is proposed, it lacks a thorough sensitivity analysis. An ablation study on the necessity and frequency of this update is needed to understand its practical contribution.
4. The paper's narrative flow could be improved. The separation of the methodology (Section 4) from its theoretical justification (Section 5) can be disruptive; integrating the theory more directly into the method's description could make the core ideas clearer.

---

> ### Author Rebuttal · Authors · 2025-07-29
>
> We are grateful for the constructive feedback and insightful comments. Please find our answers below.
>
> # The paper lacks a clear, explicit description of the mechanism used to enforce the subspace constraint on the projection matrices.
>
> We thank the reviewer for pointing this out. In practice we enforce the constraint with an inexpensive matrix multiplication:
>
> $W_{projected}  = WUU_k$.
>
> We will clearly mention this in the revision.
>
> # Ablation with $k$
>
> In our experiments, we observed that compression ratios as high as 200× can be achieved under certain hyperparameter configurations with only minimal performance degradation. However, we chose to use a more conservative 100× compression (with $k = 40$) as the default and gain consistent performance across a variety of setups (different datasets, model sizes etc.), showcasing its robustness. This choice also supports our practical goal of enabling pipeline-parallel training over internet-grade bandwidth. In response to the reviewer’s suggestion, we now include an ablation study that varies $k$ (and thus the compression ratio). The results, shown below, are for a 1B model trained on BookCorpus. We thank the reviewer for this insightful suggestion.
>
> | Method                   | Perplexity | Compression|
> |-|-|--|
> | Centralized              | 18.35      | - |
> | Decentralized (K = 40)   | 17.63      | 100x |
> | Decentralized (K = 20)   | 18.31      | 200x |
> | Decentralized (K = 10)   | 20.28      | 400x |
> | Decentralized (K = 5)    | 34.81       | 800x |
>
>
> # While a mechanism to update the shared subspace using the Grassmann manifold is proposed, it lacks a thorough sensitivity analysis.
>
> We thank the reviewer for pointing this out. Our observation is that the grassman updates matter towards mid/end training (we have provided an ablation in Fig.14), and very infrequent updates (every 500th iteration) are enough. We would add a more thorough ablation in the supplementary.
>
> # The paper's narrative flow could be improved. The separation of the methodology (Section 4) from its theoretical justification (Section 5) can be disruptive; integrating the theory more directly into the method's description could make the core ideas clearer.
>
> We appreciate the reviewer’s input. We intentionally separated Section 4 (method) from Section 5 (theory) so that readers can first grasp the overall procedure without being slowed by mathematical details. In the camera‑ready, we will improve the flow by adding forward pointers in Section 4 (e.g., “See Sec. 5.2 for the formal proof of Step 3”) and brief intuitive summaries inside Section 5. This integrates the theoretical justification more tightly with the algorithm while still keeping the main text accessible to readers who prefer a math‑light first pass.
>
> # Could Subspace Networks be combined with methods that induce low-rank structure in self-attention matrices or with other techniques used in DDP to achieve even greater communication savings?
>
> This is a very insightful question. Our subspace compression approach is indeed orthogonal to, and compatible with, other compression techniques, including those used in data-parallel (DDP). To demonstrate this, we conducted an experiment using a hybrid setup that combines DDP and pipeline-parallel PP training. In this configuration:
>
> 1. We apply low-rank compression and top-K compression to weight gradients to mimic typical DDP compression strategies.
>
> 2. Simultaneously, we apply subspace compression across the pipeline-parallel segments.
>
> We train a 1B model in this setup with 80Mbps links. This combined setup yields strong end-to-end communication savings while preserving convergence as shown in below results.
>
> | Method                                                | TPS | Perplexity @ 1k iters ↓ |
> |-|-|-|
> | Top‑K (weight gradients)                               | 32  | 52.98                    |
> | Top‑K (weight gradients)  + Ours (activations and activation gradients)    | 599 | 52.45                    |
> | PowerSGD (weight gradients)                            | 29  | 31.18                    |
> | PowerSGD (weight gradients)  + Ours (activations and activation gradients) | 532 | 30.26                    |
>
>  We also include results with Swarm parallelism [1], where our method is layered on top of standard Swarm compression (e.g., quantization).
>
> | Method                    | Loss (after 12 hours) | TPS ↑   |
> |---------------------------|------------------------|--------|
> | Swarm  | 6.05                   | 1,502  |
> | Swarm + our compression  | 4.28                   | 17,072 |
>
> These results highlight that Subspace Networks can be composed with other compression schemes (e.g., low-rank, quantized gradient exchange) to enable efficient training across heterogeneous environments with varying bandwidth constraints.
>
> We thank the reviewer for this excellent suggestion and will expand on these findings in the supplementary material.
>
> # How does this constrained optimum relate to the stationary points of the original, unconstrained problem? Could the subspace constraint potentially prevent the model from reaching a better (e.g., lower loss) stationary point that exists outside the predefined subspace?
>
> This is an intriguing question. Our central argument for the overlap, and comparable performance, between the constrained and unconstrained optima is that the final solutions for $W_{p1}$  and $W_{p2}$ already reside in the low‑rank (constrained) sub‑space. We support this claim theoretically (Theorem E.2, G.1) and  empirically: as shown in Fig. 16, we analyzed the rank distributions of the projection matrices in multiple frontier open‑source models (DeepSeek, Llama, Qwen, and OLMo, spanning 4 B to 70 B parameters) and found them all to be highly rank‑deficient.
>
> Because the true solution lies within the constrained space, projected gradient descent naturally converges to a first‑order stationary point that contains a feasible low-rank solution. While it is conceivable that alternative training strategies could discover higher‑rank optima with lower loss, our experiments and every set of publicly available weights we have examined indicate that state‑of‑the‑art models consistently converge to solutions inside the proposed low‑rank manifold.
>
>
> # In Figure 4, the 16-layer model appears to reach a similar training loss as the 8-layer model in much fewer training time.
>
> This is an intriguing question and we thank the reviewer  for noticing this. Although the 16‑layer model requires more computation, (which is visible by lower TPS in Fig. 4), it also reduces the loss more quickly per training iteration. Similar behaviour can be seen in the publicly available OLMo training curves. We invite the reviewer to inspect the public WandB logs for OLMo, which exhibit the same trend. (We omit direct links here to comply with NeurIPS rules.)  Because the larger model needs fewer iterations to reach a given loss, its wall‑clock time to a similar loss can be shorter despite the higher per‑step compute cost.
>
> # Minor suggestions:
>
> Thank you for pointing these out. We will address all of the above.
>
> [1] - Ryabinin et al. -  Swarm Parallalism

---

> > ### Comment · Reviewer_VZDz · 2025-08-04
> >
> > Thanks for your detailed response. I think most of my concerns are addressed. I will adjust my score accordingly.

---

### Official Review · Reviewer_HSht · 2025-07-02

**Clarity:** 3
**Significance:** 2
**Originality:** 2
**Rating:** 5
**Confidence:** 3

**Summary:**

The paper focuses on distributed training using model parallelism in bandwidth-constrained environments (e.g., Internet/WAN).
Observing that activations and gradients mostly reside in a low-dimensional subspace, the authors project both (forward and backward) onto a low-rank basis for nearly lossless compression. As a result, their method achieves up to 99 % compression without degrading convergence. The improvement delivers a comparable time-to-accuracy under 80 Mbps links to that measured in datacenter-grade networks (>16 Gbps).

**Questions:**

- Although Section 6.5 addresses robustness, a broader evaluation would be helpful. Is rank collapse consistently observed across different models? Even if so, might the optimal rank dimension vary with model type, architecture, size, or training scheme (pre-training vs. fine-tuning)?
- If the rank does vary, how can it be detected? For instance, the $k$ parameter seems almost randomly chosen, are there principled ways to adapt it to each model?
- How do the training curves change as a function of $k$?
- What does the loss curve look like if training continues far longer than in Figures 2–5 (until full convergence)? I’m wondering that the compression does not affect the final convergence.
- How does the loss curve compare with other recent works that compress activations and gradients for MP? What advantages does this method offer over those approaches? What are the key differences from other low-rank projection techniques?
- What is the overhead of the subspace-projection and full-reconstruction steps?

Typo

L311: 13x -> 13$\times$

**Ethical Concerns:**

["NO or VERY MINOR ethics concerns only"]

**Final Justification:**

I have raised my score from 4 to 5 (“Accept”). The paper presents a simple yet powerful approach, firmly grounded in a solid theoretical foundation, that I believe will have substantial impact.

**Limitations:**

yes

**Paper Formatting Concerns:**

No concerns.

**Quality:**

2

**Strengths And Weaknesses:**

**Strengths**
- The paper provides a thorough theoretical background based on mathematical formulations, establishing a clear link between the proposed algorithm to its experiment.
- The paper shows impressive compression, 99\% reduction without degrading convergence, through nearly lossless subspace projection.
- The algorithm is simple (though not trivial) and effective. It appears generally applicable to all kinds of LLMs including transformer blocks, and is highly suitable for geo-distributed deployments aimed at cost-effective LLM serving or training.

**Weakness**
- **Novelty**: The motivation is quite conventional. Although the algorithm is theoretically concrete and effective, both the motivation and the way it is addressed lack novelty. As mentioned in Section 4.1, the low-rank property is a well-known phenomenon, and a line of prior work already compresses parameters (weights, activations, or gradients) using low-rank projection.
- **Lack of evaluation**: Consistent with the point above, numerous works with similar motivations and solutions exist, yet the evaluation does not cover them thoroughly. While the experiments demonstrate the algorithm’s own merits, the paper doesn't include comparisons with other MP or DP methods. Similarly, there are insufficient comparisons with alternative compression techniques, including other low-rank projection approaches.

---

> ### Author Rebuttal · Authors · 2025-07-29
>
> We thank the reviewer for the insightful comments and feedback. We address the reviewer's concerns below.
>
> # Novelty,  the low-rank property is a well-known phenomenon, prior work already uses low-rank projection.
>
> We thank the reviewer for their feedback and would like to clarify key points that distinguishes our contribution.
>
> **1. Low‑rank structure of  activations is *not* a well known property**
>
> While the community has long recognized that **weight matrices and their gradients often exhibit low effective rank** (enabling techniques such as LoRA, PowerSGD) **the same assumption does not hold for activations or activation gradients**. In transformers, activations are inflated in rank because PE and TE are re‑added at every layer. This resulted in an obscure view in earlier studies, leading them to report  accuracy drops when activations were compressed beyond ≈10%, concluding that aggressive activation compression is infeasible [1,2]. Another contributing factor to this is the exponential error accumulation (Theorem D.1).
>
> **2. Our key insight: remove embedding components, reveal a hidden low‑rank subspace**
>
> **We show that once PE and TE contributions are removed, the residual activations lie in the span of the layer’s output‑projection matrices**, which is a provably low‑rank subspace.
>
> Concretely: 1) PE can be subtracted inside each block because they are deterministic. 2) TE are batch‑dependent and high‑rank. We decompose them into a fixed high‑rank component and a learnable low‑rank component that can be compressed efficiently.
>
> **This decomposition exposes a low‑rank structure in activations and their gradients that, to our knowledge, has not been identified or exploited in prior work.**
>
> **3. Practical impact: efficient pipeline‑parallel training**
>
> By isolating the compressible subspace, we achieve activation and activation‑gradient compression well beyond the previous rates,  without measurable loss in accuracy, enabling end‑to‑end pipeline‑parallel training with dramatically lower bandwidth. Previous methods focused on weight/gradient compression could not unlock this regime. **Also the network provably converges to a feasible solution with constrained optimization**
>
> We hope this clarifies that our contribution goes beyond applying a “standard” low‑rank trick: it uncovers and leverages a previously overlooked low‑rank structure in activations.
>
> #  Comparisons with other MP or DP methods.
>
> **1. Comparison against other DP methods**
>
> Because our algorithm is tailored to MP training, it is not directly comparable to DP approaches. DP assumes each full model replica lives on a high‑capacity cluster, which is less restrictive than splitting the computational graph across the internet, which is the setting we target. Existing decentralized or communication‑efficient DP methods address slow bandwidth between clusters, not between individual model layers, making that problem comparatively easier.
>
> **Our method is therefore orthogonal and complementary to DP communication‑reduction techniques and can be layered on top of them.** To illustrate this, we present an additional experiment below: we applied top‑k and low‑rank compression to the weight gradients (replicating a DP scenario) while simultaneously using our scheme to compress activations and activation gradients. Under an 80 Mbps link between devices, we achieved a substantial increase in TPS with no loss of convergence.
>
> | Method                                                | TPS | Perplexity @ 1k iters ↓ |
> |-|-|-|
> | Top‑K (weight gradients)                               | 32  | 52.98                    |
> | Top‑K (weight gradients)  + Ours (activations and activation gradients)    | 599 | 52.45                    |
> | PowerSGD (weight gradients)                            | 29  | 31.18                    |
> | PowerSGD (weight gradients)  + Ours (activations and activation gradients) | 532 | 30.26                    |
>
> **2. Comparison against other MP methods**
>
> As noted in our previous response, activation compression in model‑parallel (MP) pre‑training has generally been considered harmful to convergence, so we are not aware of any published method that successfully compresses both activations and activation gradients during pre‑training.
>
> The recent study [2] (which does not propose a new algorithm) benchmarks 8‑bit/4‑bit quantization and top‑K sparsification. We already include these baselines, and a vanilla low‑rank SVD variant, in Fig. 6, where our method significantly outperforms them. We would also like to add a note on why our method outperforms low-rank SVD methods.
>
> *Why our algorithm outperforms low‑rank SVD?*
>
> 1. Embedding removal & constrained factorization of the projection matrices expose a strictly lower‑rank subspace; naïve SVD sees the full high‑rank tensor.
>
>
> 2. No per‑batch SVD: we replace the O(d³) decomposition with a single matrix multiplication derived from our constrained optimization, yielding substantial runtime savings.
>
> 3. Convergence guarantees as our method can be formulated as a projected gradient descent problem.
>
> Further, [3] proposes compressing activation deltas during fine‑tuning, but this technique is infeasible for large‑language‑model (LLM) pre‑training: 1) **Memory overhead.** It must store the full activations for every training example, which is infeasible at LLM scale. 2) **Requirement for multi epoch training.** Error elimination requires many epochs, and the method fails to converge in the few‑epoch regimes typical of large‑scale training, where workloads often cover less than a single pass over the dataset. A different research thread ([4,5,6]) focuses on memory efficiency by compressing activations within each module and decompressing them later to compute weight gradients. Because they leave the inter‑device traffic unchanged, these methods do not address communication efficiency and are therefore not comparable to ours.
>
>
> We did identify a very recent workshop paper [7] that uses column masking to compress activations. However, it achieves only up to 10× compression, whereas our technique reaches 100×. For completeness we have added a comparison: their convergence degrades sharply at an aggressive compression rate, while ours remains stable. We will add this to supplementary.
>
> | Method                | Perplexity 	↓ |
> |-----------------------|------------|
> | [7] (100× compression) | 147.15     |
> | Ours (100× compression)| 17.63      |
>
> **If the reviewer is aware of additional MP activation‑compression algorithms, we would be grateful for citations. We will try our best to provide ablation tables for such methods in the discussion period and include in the supplementary material.**
>
> We further conducted an experiment with  Swarm parallelism [8]. Swarm’s  primary goals are fault tolerance and dynamic volunteer participation rather than communication efficiency. Swarm implementation relies on standard activation and gradient quantization for compression. Our technique is therefore is orthogonal to swarm too and can be layered on top of Swarm to further reduce bandwidth. To demonstrate this, we have conducted an additional experiment that applies our compression within a Swarm setting; the results are provided below.
>
> | Method                    | Loss (after 12 hours) | TPS ↑   |
> |---------------------------|------------------------|--------|
> | Swarm  | 6.05                   | 1,502  |
> | Swarm + our compression  | 4.28                   | 17,072 |
>
> #  Is rank collapse consistently observed across different models? Even if so, might the optimal rank dimension vary with model type, architecture, size, or training scheme?
>
> Yes. As shown in Fig. 16, we analyzed the rank distributions of the projection matrices in multiple frontier open‑source models (DeepSeek, Llama, Qwen, and OLMo, spanning 4 B to 70 B parameters) and found them all to be highly rank‑deficient.
>
> In our experiments, we were able to achieve compression ratios of up to 200× under certain hyperparameter settings, with minimal impact on performance. However, we adopt a more conservative 100× compression (using $k=40$) as the default to ensure stable performance across all of our configurations. Note that we use the gain consistent performance with this value across all our ablations (different datasets, model sizes etc.). This setting also aligns with our practical objective: enabling pipeline-parallel training over internet-grade connections. In response to the reviewer’s suggestion, we now include an ablation study varying $k$ (and thus the compression ratio). Results are reported below for a 1B model trained on BookCorpus. We sincerely thank the reviewer for this valuable feedback.
>
> | Method                   | Perplexity | Compression|
> |-|-|--|
> | Centralized              | 18.35      | - |
> | Decentralized (K = 40)   | 17.63      | 100x |
> | Decentralized (K = 20)   | 18.31      | 200x |
> | Decentralized (K = 10)   | 20.28      | 400x |
> | Decentralized (K = 5)    | 34.81       | 800x |
>
> # What does the loss curve look like if training continues until full convergence?
>
> We have already trained models to full convergence using the compute-optimal 1:20 model-to-token ratio from the Chinchilla scaling law [9] in Appendix F. We achieve even better convergence than centralized models.
>
> # What is the overhead of the subspace-projection?
>
> We have already provided these measurements in Appendix L.1. The overhead is approximately 1%, indicating a negligible computational burden.
>
>
> [1] - Bian et al. - Does Compressing activations help model parallel training?
>
> [2] - Rudakov et al. - Activations and Gradients Compression for Model-Parallel Training
>
> [3] - AQ-SGD
>
> [4] - Shamshoum et al. - CompAct
>
> [5] - Chen et al. - ActNN
>
> [6] - Evans et al. AC-GC
>
> [7] - Ramasinghe et al. - Beyond Top-K
>
> [8]- Ryabinin et al. - Swarm Parallalism.
>
> [9] - Chinchilla

---

> > ### Comment · Reviewer_HSht · 2025-08-04
> >
> > Thank you to the authors for the kind and thoughtful response. Your reply has addressed most of my concerns, and in particular, it has clarified the distinctions from other approaches and made the contributions more evident.

---

### Official Review · Reviewer_ppVQ · 2025-07-02

**Clarity:** 4
**Significance:** 3
**Originality:** 4
**Rating:** 5
**Confidence:** 3

**Summary:**

The paper introduces a novel compression algorithm for model parallelism in decentralized training. By constraining projection matrices and activations to a low-dimensional subspace learned via Grassmann manifold updates, the method enables exact reconstruction of activations and gradients across layers. This yields up to 100× communication reduction with negligible compute/memory overhead, allowing billion-parameter transformer models to converge on 80 Mbps consumer-grade links as fast as on 100 Gbps datacenter networks

**Questions:**

-  The paper says the method has little extra computation or memory cost. It would provide more clarity if the measurements of the computation and memory is clearly stated in the start of section 6
- The comparison to a centralized method is too brief. As the performance of the proposed method is very similar to the centralized method, can you provide more background introduction of the centralized method and explanation to understand this?
- The words in the plot is too small.
- In the figures in Appendix I, why the decentralized with no compressed is not visible when the x-axis is training steps? Would the loss be similar as the proposed method regarding the training steps?
- The paper uses the Frobenius norm to measure distance between subspaces. What is the intuition behind the use of F norm?

**Ethical Concerns:**

["NO or VERY MINOR ethics concerns only"]

**Final Justification:**

The paper presents solid theoretical analysis and comprehensive empirical experiments. During the rebuttal stage, the authors provided extensive ablation studies that were missing in the original submission. Given these improvements, I maintain my recommendation to accept the paper.

**Limitations:**

yes

**Paper Formatting Concerns:**

No formatting issues observed

**Quality:**

3

**Strengths And Weaknesses:**

**Strengths**
- The paper provides a novel and effective method for the decentralized training, and the method is validated through thorough experiments and analyses.
- The writing provides a clear, concise motivation for decentralized model parallelism and its communication challenges.
- The linkage between empirical observations (rank collapse) and theoretical analysis (Statements 5.2–5.3) is well articulated.
- The mathematical treatment of the algorithm is beautiful.

**Weaknesses**
- More ablations would improve the quality of paper.
  - The experiments involve training across different locations, but there’s no study on how the number of machines affects training speed or results.
  - The paper doesn’t explore how the initial subspace  setup of $U_k$, especially the choice of $k$. affects training. Analyses on how the subspaces drift in the experiments (as mentioned in section 4.5) is helpful.
- The proposed method’s main advantage is that it enables training over low-bandwidth connections. What isn’t clear is exactly at what bandwidth threshold it starts to outperform other approaches—answering this would require comparing multiple methods across a range of bandwidths.

---

> ### Author Rebuttal · Authors · 2025-07-29
>
> We thank the reviewer for the positive comments and intriguing questions. Please find our answers below.
>
> #  It would provide more clarity if the measurements of the computation and memory is clearly stated in the start of section 6
>
> Thank you for the helpful suggestion. We have included detailed memory and computation analysis in Supplementary Sections K and L (see Tables 3 and 4) and show that ours incur extremely minimal overhead. As per the reviewer’s recommendation, we will also add a brief summary of these measurements at the beginning of Section 6 to improve clarity.
>
> # Can you provide more background introduction of the centralized method and explanation
>
> We appreciate the reviewer’s concern. In our setting, the centralized model refers to the same architecture as the decentralized version, but deployed on a high-performance GPU cluster, where each transformer block (residing on a separate GPU) is connected to the others via high-bandwidth, low-latency links. This setup allows near-seamless communication between layers.
> In contrast, our decentralized model places each transformer block on a node connected via low-bandwidth links, such as internet-grade connections. This introduces significant communication constraints, particularly for activations and gradients, which our method is specifically designed to address.
>
> The strong alignment in performance between these two regimes highlights the effectiveness of our compression scheme in bridging the performance gap caused by bandwidth limitations. We will expand this explanation in the revised version to clarify the distinction and significance of the centralized baseline.
>
> # The words in the plot is too small.
>
> Thank you for pointing this out. We will fix this.
>
> # Why the decentralized with no compressed is not visible when the x-axis is training steps? Would the loss be similar as the proposed method regarding the training steps?
>
> The **decentralized uncompressed model** exhibits identical training dynamics to the **centralized model** when plotted against training steps, as the architecture and optimization procedure are the same. The only difference lies in the communication bandwidth between GPUs. As a result, its loss curve completely overlaps with that of the centralized model in figures where the x-axis is training steps, which is why it is not separately visible.
>
> However, it is important to note that, as shown, the tokens-per-second (TPS) of the decentralized uncompressed model is significantly lower due to the high communication cost over low-bandwidth links. This results in **much slower convergence** in wall-clock time, which is where our proposed method provides a clear advantage by enabling efficient training under bandwidth constraints.
>
> # What is the intuition behind the use of F norm?
>
> This is indeed a great question. Below we summarize the rationale behind this choice.
>
> 1. **Relation to principal angles**
>    For two $k$‑dimensional sub‑spaces $\mathcal U,\mathcal V $ with orthonormal bases $U$ and $V$,
>
> $$d_{F}(\mathcal U,\mathcal V)
>        =(|UU^{\top}-VV^{\top})|_{F}
>        =\sqrt{2\sum \sin^{2}\theta_i},$$
>
>    where $\theta_i$ are the principal angles.  Therefore, minimising $d_F$ therefore minimises the **average squared mis‑alignment** across all $k$ directions.
>
> 2. **Rotational invariance**
>    Because the metric relies only on the projection operators $UU^{\\top}$ and $VV^{\\top}$, any orthonormal basis that spans the same sub‑space yields **exactly the same value**.  This ensures the distance is defined for sub‑spaces themselves, not for a particular choice of basis.
>
> 3. **Average‑case discrepancy**  (instead of worst‑case)
>    While the spectral norm captures only the largest principal angle (worst‑case deviation), the Frobenius norm **sums over all angles**, giving an overall or average measure of how much the two sub‑spaces differ. This is the relevant notion of closeness when we care about preserving information in every direction, not just the single worst one.
>
> # The experiments involve training across different locations, but there’s no study on how the number of machines affects training speed or results.
>
> Thank you for raising this point. To address it fully, could you please clarify what configuration you have in mind? Specifically, are you referring to increasing the number of nodes in a data‑parallel (DDP) setup? In that case, the results would be the same since that would be akin to training with a larger batch size. If the reviewer has a different setup in mind, we would like to  respectfully ask for more information so we can run the appropriate ablations and report their effects on throughput and convergence.
>
> # How the initial subspace setup of $U_k$ affects training.
>
> We initialise $U_k$​ by drawing an isotropic Gaussian matrix and taking its top‑k principal components via SVD. After orthonormalising $U_k$​, the specific directions and the initial standard deviation no longer influence convergence.
>
> # How does $k$ affect training.
>
> In our experiments, under specific hyperparameters, we achieved compression ratios as high as 200× without much performance degradation. Nonetheless, we used a more conservative 100× ratio ($k=40$) by default for stable performance across setups. This ratio also meets our practical goal: training in a pipeline-parallel setup even when nodes are connected via internet-grade networks. To address the reviewer’s request, we now include an ablation study varying k (and therefore the compression ratio). The results are shown below for a 1B model over BookCorpus. We thank the reviewer for the suggestion.
>
> | Method                   | Perplexity | Compression|
> |--------------------------|------------|------------|
> | Centralized              | 18.35      | - |
> | Decentralized (K = 40)   | 17.63      | 100x |
> | Decentralized (K = 20)   | 18.31      | 200x |
> | Decentralized (K = 10)   | 20.28      | 400x |
> | Decentralized (K = 5)    | 34.81       | 800x |
>
> We agree that exploring the true upper limits and deriving more intuitive heuristics (potentially tied to model architecture or task requirements) is an exciting direction for future research.
>
> # Analyses on how the subspaces drift in the experiments  is helpful.
>
> We agree that reporting subspace drift would be valuable, and we will add these statistics in the revision.
>
> # What isn’t clear is exactly at what bandwidth threshold it starts to outperform other approaches—answering this would require comparing multiple methods across a range of bandwidths
>
> Thank you for this important question. As requested, we provide TPS comparing our method with alternative approaches across a range of bandwidths.
>
> We observe that top‑k sparsification allows up to a 10× compression and 8‑bit quantization enables around 4× compression before significant degradation in convergence. Note that these findings are consistent with those reported in [1]. Accordingly, we fix those compression ratios for the baselines. We exclude SVD‑based low‑rank method as it is extremely slow.
>
> Our method achieves up to 100× compression without compromising convergence. This allows it to outperform other methods even when the available bandwidth drops to 80Mbps. The following results (to be included in the supplementary) show that our approach demonstrates a consistent TPS compared to  other methods across the range of bandwidths, making it highly practical for pipeline‑parallel training over internet-grade connections.
>
> We appreciate the reviewer’s suggestion and will include the table and performance plots in the supplementary.
>
> | Method       | TPS @ 100 Gbps | TPS @ 1 Gbps | TPS @ 800 Mbps | TPS @ 500 Mbps | TPS @ 80 Mbps |
> |--------------|----------------|--------------|----------------|----------------|----------------|
> | Baseline     | 602            | 378          | 333            | 222            | 36             |
> | Top-K        | 612            | 532          | 521            | 451            | 129            |
> | Quantization | 608            | 499          | 487            | 415            | 83             |
> | Ours         | 624            | 612          | 610            | 603            | 592            |
>
> [1] - Rudakov et al. - Activations and Gradients Compression for Model-Parallel Training

---

> > ### Comment · Reviewer_ppVQ · 2025-08-06
> > **Response**
> >
> > Thanks to the authors for their response. It has effectively resolved my questions.
> >
> > To clarify a point that seemed to cause some confusion:
> > > To address it fully, could you please clarify what configuration you have in mind? Specifically, are you referring to increasing the number of nodes in a data‑parallel (DDP) setup?
> >
> > I was referring to line 63, where the authors mention splitting the layers across four different geographical regions with 8 instances. My concern was that the number of geographical regions could significantly influence the results in practice.
> >
> > However, I am already satisfied with the response, even without additional results on this specific aspect.

---

> ### Author Response · Authors · 2025-08-08
> **On distributing the nodes across a varying number of geographical regions.**
>
> We are thankful to the reviewer for the continued engagement and the thought provoking question.
>
> ## On distributing the nodes across a varying number of geographical regions
>
> In the **bandwidth-limited, steady-state regime** where (i) per-layer compute and **intra-region** transfers costs are negligible, and (ii) throughput is limited by **inter-region** links, the **slowest inter-region link** bottlenecks and sets tokens-per-second (TPS). Once the pipeline is saturated, **changing the number of regions \(d\)** does **not** meaningfully change TPS assuming a standard number of micro batches.
>
> ---
>
> ## Analysis
>
> Let:
> - $n$: total layers, partitioned across \(d\) regions
> - $p$: **bytes in activations** that must cross a region boundary (after compressing activations & activation gradients)
> - $B_i$: bandwidth of inter-region link $i$; let $B_{\\min}=\\min_i B_i$
> - $m$: number of microbatches (pipeline chunks)
> - $\alpha$: scheduling constant ($\\alpha \\approx 1 $ for GPipe; $\\alpha \\approx 2 $ for 1F1B)
>
> With communication time $\\gg$ compute time, the stage service time is dominated by the worst boundary:
> $$
> T_{\text{stage}} \approx \max_i \frac{p}{B_i} = \frac{p}{B_{\min}}.
> $$
> In steady state (pipeline filled),
>
> $$
> \\text{TPS} \\approx  \\frac{B_{\\min}}{p} \\times \\frac{m}{m+\\alpha(d-1)}.
> $$
>
> Typically **$m \\gg d$**, yielding,
> $$
> \boxed{\text{TPS} \approx \frac{B_{\min}}{p}\ \ \text{(independent of } d\text{)}}.
> $$
>
> ---
> However, If  $m$ is too small, more regions increase bubble overhead. Nonetheless, in typical pipeline parallel settings, $m$ is chosen to saturate the pipeline.
>
> Therefore, under the stated assumptions (bandwidth-dominated, saturated pipeline, isolated links), **TPS is governed by $B_{\\min}/p$**. Thus, **the number of geographical regions $d$** is **not** a primary determinant of throughput. We will clarify these assumptions and include the above analysis in the appendix to make this explicit and try our best to add an ablation. Thank you for the valuable suggestion.

---

### Official Review · Reviewer_3Fcv · 2025-07-03

**Clarity:** 4
**Significance:** 4
**Originality:** 4
**Rating:** 5
**Confidence:** 3

**Summary:**

The work proposes exploiting the observation that the weight gradients of projection matrices in a transformer block reside in a low rank space after training. The authors explicitly constrain the projection matrices to span a low rank space by matrix decomposition. By compressing the activations and weight gradients, the communication overhead is significantly reduced, allowing for model parallelism schemes where network bandwidth is limited.

**Questions:**

My primary request is to include baseline methods that cover pipeline parallelism or existing model parallelism strategies (see weakness). The work is technically sound and doesn’t sacrifice model convergence or performance. However, it is hard to contextualize the results within the broader literature without these points of comparison. With the inclusion of additional baselines, I am willing to raise my score.

**Ethical Concerns:**

["NO or VERY MINOR ethics concerns only"]

**Final Justification:**

My main concern revolved around the baseline to compare to, most notably to pipeline parallelism, which I felt the authors adequately addressed in their rebuttal. To ask the authors for another MP activation-compression algorithm or PP algorithm when it does not exist yet, would be out of scope. I think the work stands alone regardless and the baselines in the paper and response are sufficient. As a result, I will be raising my score from a 4 to a 5.

**Limitations:**

yes

**Quality:**

3

**Strengths And Weaknesses:**

Strengths:
- The proposed method seems to be theoretically sound and well motivated.
- The theoretical insights are empirically verified, notably in 6.4 (fig 5) where the decentralized compressed model achieves similar convergence to the centralized model.
- The provided experiments, especially with cross-regional compute instances, are compelling .

Weaknesses:
- The baseline methods are not representative of other communication efficient training schemes. The work compares to lossy compression schemes but not to other pipeline parallelism methods (e.g., Swarm parallelism).

---

> ### Author Rebuttal · Authors · 2025-07-29
>
> Thank you for the encouraging comments and constructive feedback. We address each of the reviewer’s specific concerns below.
>
> # My primary request is to include baseline methods that cover pipeline parallelism or existing model parallelism strategies (swarm parallalism)
>
> We thank the reviewer for the suggestion. Swarm implementation relies on standard activation and gradient quantization, and multiple node integration for computation pipes, for compression and increased throughput. Our technique is therefore orthogonal and can be layered on top of Swarm to further reduce bandwidth. To demonstrate this complementarity and address the reviewer’s concern, we have conducted an additional experiment that applies our compression within a Swarm setting; we train a 1B parameter model in swarm, with and without our compression. Results are summarized below and we will add the loss curve in the revised supplementary. As illustrated in, with our compression, swarm achieves a remarkable >11x throughput gain.
>
> | Method                    | Loss (after 12 hours) | TPS ↑   |
> |---------------------------|------------------------|--------|
> | Swarm  | 6.05                   | 1,502  |
> | Swarm + our compression  | 4.28                   | 17,072 |
>
> Please note that activation and activation‑gradient compression during model‑parallel pre‑training has long been viewed as harmful to convergence, making it difficult to find suitable baselines. The recent survey paper [1], which proposes no new algorithm, evaluates 8‑bit / 4‑bit quantization and top‑k sparsification for MP. We already include these baselines, along with a vanilla low‑rank SVD variant, in Fig. 6; our method consistently outperforms them.
>
> [2] introduces an activation‑delta compression scheme for fine‑tuning, but it is impractical for LLM pre‑training for two reasons. (1) **Memory overhead**: the method must cache full activations for every training sample, which is infeasible at LLM scale. (2) Multi‑epoch dependence: the accumulated error is only removed after many epochs, so the approach fails to converge in the few‑epoch regimes typical of large‑scale training, where workloads rarely exceed a single pass over the data.
> A different research thread ([3,4,5]) focuses on memory efficiency by compressing activations within each module and decompressing them later to compute weight gradients. Because they leave the inter‑device traffic unchanged, these methods do not address communication efficiency and are therefore not comparable to ours.
>
> Upon further inspection, we did find a very recent workshop paper [6] that applies column masking to compress activations, but it achieves at most a 10 × reduction. In contrast, our technique delivers up to 100 × compression. We ran an experiment to validate the efficacy of our method using a 1B model on BookCorpus. The results are shown below after 5k iterations. Note  that their method’s convergence is  notably inferior to ours at 100x compression rate.
>
> | Method                | Perplexity 	↓ |
> |-----------------------|------------|
> | [6] (100× compression) | 147.15     |
> | Ours (100× compression)| 17.63      |
>
> **If the reviewer is aware of any other MP activation‑compression algorithms, we would greatly appreciate the citations. We will do our best to add ablation results for those methods during the discussion period and include them in the supplementary material.**
>
> [1] - Rudakov et al. - Activations and Gradients Compression for Model-Parallel Training
>
> [2] - Wang et al. AQ- SGD
>
> [3] - Shamshoum et al. - CompAct
>
> [4] - Chen et al. - ActNN
>
> [5] - Evans et al.  AC-GC
>
> [6] - Ramasinghe et al. Beyond Top-K

---

> > ### Comment · Reviewer_3Fcv · 2025-08-06
> > **Response to the authors**
> >
> > Thank you for taking the time to address my concerns and to clarify where the work sits in the broader literature. I am not aware of other MP activation-compression algorithms, and will be raising my score.

---

### Comment · Area_Chair_5sCD · 2025-08-04
**Gentle Reminder: Reviewer Response to Rebuttal Needed**

Dear reviewers,

Thank you for your valuable time and expertise in reviewing. The author rebuttal phase is nearing its close, and we kindly request your prompt attention to ensure a thorough discussion.

**The discussion period ends in less than 3 days (on Aug. 6, 11:59pm AOE )**. To maintain the review timeline, we ask that you:

1. Review the authors’ rebuttal,

2. Engage in any ongoing discussion with fellow reviewers/authors (if applicable),

3. Finalize your assessment.


If you have already completed this step, please disregard this reminder—we sincerely appreciate your efforts.
Your timely input is crucial to the integrity of our review process. Thank you for your collaboration!

Best regards,

AC

---

### Decision · Program_Chairs · 2025-09-17

**Decision:**

Accept (poster)

**Comment:**

This paper introduces a novel technique for compressing communication in model-parallel training by projecting activations and gradients onto a low-dimensional subspace. The method achieves a remarkable 99% lossless compression ratio, enabling highly efficient decentralized training over low-bandwidth wide-area networks while matching the performance of datacenter-grade infrastructure.

The approach is both novel and effective. The connection between the empirical observation of rank collapse and the theoretical foundation is well articulated, and the experimental results—particularly the cross-regional training simulations—are compelling. The authors have thoroughly addressed all reviewers' concerns through extensive additional experiments and clarifications. As a result, all reviewers converged to strong positive ratings. Given the strength of the methodology and author responses, I recommend acceptance.